# The HCV-Dependent Inhibition of Nrf1/ARE-Mediated Gene Expression Favours Viral Morphogenesis

**DOI:** 10.3390/v17081052

**Published:** 2025-07-28

**Authors:** Olga Szostek, Patrycja Schorsch, Daniela Bender, Mirco Glitscher, Eberhard Hildt

**Affiliations:** 1Research Group, Paul-Ehrlich-Institute, 63225 Langen, Germanydaniela.bender@pei.de (D.B.);; 2Digital Health Cluster, Hasso-Plattner-Institute, University of Potsdam, 14482 Potsdam, Germany

**Keywords:** hepatitis C virus (HCV), nuclear factor erythroid 2 related factor-1 (Nrf1), nuclear factor erythroid 2 related factor-2 (Nrf2), cholesterol, small Maf proteins (sMaf)

## Abstract

The life cycle of the hepatitis C virus (HCV) is closely linked to lipid metabolism. Recently, the stress defence transcription factor, nuclear factor erythroid 2 related factor-1 (Nrf1), has been described as a cholesterol sensor that protects the liver from excess cholesterol. Nrf1, like its homologue Nrf2, further responds to oxidative stress by binding with small Maf proteins (sMaf) to the promotor antioxidant response element (ARE). Given these facts, investigating the crosstalk between Nrf1 and HCV was a logical next step. In HCV-replicating cells, we observed reduced levels of Nrf1. Furthermore, activation of Nrf1-dependent target genes is impaired due to sMaf sequestration in replicase complexes. This results in a shortage of sMaf proteins in the nucleus, trapping Nrf1 at the replicase complexes and further limiting its function. Weakened Nrf1 activity contributes to impaired cholesterol removal, which occurs alongside an elevated intracellular cholesterol level and inhibited LXRα promoter activation. Furthermore, inhibition of Nrf1 activity correlated with a kinome profile characteristic of steatosis and enhanced inflammation—factors contributing to HCV pathogenesis. Our results indicate that activation of Nrf1-dependent target genes is impaired in HCV-positive cells. This, in turn, favours viral morphogenesis, as evidenced by enhanced replication and increased production of viral progeny.

## 1. Introduction

The hepatitis C virus (HCV) is an enveloped RNA virus classified within the Hepacivirus genus of the *Flaviviridae* family [1]. Its genome is a positive-sense RNA molecule, approximately 9.6 kilobases in length, which is translated into a single polyprotein. This polyprotein is subsequently cleaved into three structural proteins (core protein, glycoproteins E1 and E2) and seven non-structural proteins (p7, NS2, NS3, NS4A, NS4B, NS5A and NS5B) [2].

It is estimated that more than 50 million people globally are suffering from hepatitis C infection. On average, 30% of infected individuals experience the acute form of viral hepatitis, which in most cases can be cleared spontaneously. In the remaining 70% of the cases, HCV infection leads to chronic viral hepatitis C (CHC). CHC is a significant health burden worldwide, with serious implications such as liver cirrhosis, hepatocellular carcinoma, increased risk of type 2 diabetes, or hypercholesterolemia [3,4,5]. Current therapeutic approaches against HCV infection involve the use of third-generation direct-acting antivirals (DAAs), which have replaced the earlier interferon-alpha (IFN-α) and Ribavirin-based treatments [6]. Nevertheless, due to economic inequity in low-income countries, the accessibility to DAAs often remains poor alongside social awareness [7]. Therefore, the development of an effective HCV vaccine remains an ongoing challenge [8].

Infection with the hepatitis C virus (HCV) was found to have significant effects on lipid metabolism and the rearrangement of intracellular membranes. Specifically, HCV manipulates the host cell’s endoplasmic reticulum (ER)-derived membranes to facilitate its own replication. HCV replication occurs in the replicase complexes on the cytoplasmic face of the ER [9]. They consist of viral RNA, non-structural proteins and host cell factors. The precise localisation for the replication to happen is within specialised structures known as double-membrane vesicles (DMVs). Originating from the ER, these DMVs provide a suitable environment for viral RNA replication [10]. HCV assembly occurs in close proximity to the ER, in regions known as detergent-resistant membranes (DRMs) or mitochondrial-associated ER membranes (MAMs) [11]. These membrane regions have a distinctive lipid composition, being highly enriched in cholesterol and sphingolipids. A crucial link between HCV replication and assembly is established through cytosolic lipid droplets (cLDs) [9]. cLDs serve as a connecting point between viral replication sites in DMVs and assembly sites in DRMs or MAMs. They play a pivotal role in facilitating the transfer of viral components between these compartments, supporting the efficient assembly of infectious HCV particles [12]. Moreover, HCV virions are closely associated with various lipoproteins and apolipoproteins, resulting in the creation of lipoviroparticles (LVPs). Virion assembly, infectivity and entry strongly depend on the virus association with lipoproteins. The manipulation of lipid metabolism and the rearrangement of intracellular membranes by HCV are critical for its successful replication and assembly within the host cell [13].

Hepatitis C virus infection is known to induce oxidative stress and to disrupt redox homeostasis, leading to cellular damage, altered cellular metabolism and impaired liver regeneration [14,15]. A key protective mechanism against oxidative stress is the Nrf2/Keap1 signalling pathway. Nuclear factor erythroid 2 related factor-2 (Nrf2) is a transcription factor that plays a crucial role in activating the expression of cytoprotective genes [16]. Under normal physiological conditions, Nrf2 is bound to its inhibitor Kelch-like ECH-associated protein 1 (Keap1). However, upon activation, the Nrf2-Keap1 complex dissociates, allowing translocation of Nrf2 into the nucleus. Inside the nucleus, Nrf2 forms a heterodimer with small Maf proteins (sMafs) and binds to specific sequences known as antioxidant response elements (AREs) or electrophile response elements (EpREs). This binding activates the expression of cytoprotective genes [17,18,19,20]. In the context of HCV replication, impaired Nrf2/ARE signalling has been observed in the infected cells. This impairment is attributed to the withdrawal of sMaf proteins from the nucleus as they bind to NS3, an integral component of the HCV replicase complex, on the cytoplasmic face of the ER. Upon binding to NS3 at the replicase complexes, sMafs are withdrawn from the nucleus to the surface of the ER. The dislocation of sMaf from the nucleus to the surface of the ER has two consequences: (i) it leads to a lack of sMaf proteins in the nucleus, and (ii) it prevents Nrf2 from entry into the nucleus, as the released Nrf2 is trapped and sequestered by the ER-localised sMafs. This prevents the Nrf2-mediated induction of cytoprotective ARE-dependent genes. Consequently, the reactive oxygen species (ROS) within HCV-infected cells are not detoxified, and the ROS levels remain elevated [21]. Apart from its relevance for HCV-associated pathogenesis, the elevated ROS level is crucial for the HCV life cycle by the activation of autophagy. On the one hand, autophagy represents a cellular defence mechanism; on the other hand, it is required for the MVB-dependent release of HCV in the form of exosomes [19].

Another factor involved in maintaining the cellular redox homeostasis is the nuclear factor erythroid 2 related factor-1 (Nrf1; NFE2L1), a ubiquitously expressed transcription factor, belonging to the Cap′N′Collar family [22]. In its unprocessed form, full-length Nrf1 is bound to the ER membranes. Upon stimulation, the inactive 120 kDa N-glycoprotein undergoes deglycosylation by N-glycanase 1 (NGLY1) and cleavage by the ubiquitin-directed endoprotease DNA damage inducible 1 homologue 2 (DDI2), resulting in the activation of the p95-Nrf1 isoform [23,24]. Subsequently, Nrf1 interacts with the endoplasmic reticulum-associated degradation (ERAD) machinery mediated by the proteasome [25], leading to its cleavage [22]. This releases Nrf1 from the ER, allowing its translocation to the nucleus. Upon translocation, Nrf1 undergoes further processing and cleavage, generating multiple truncated isoforms. These isoforms exhibit distinct activities and are responsible for regulating specific subsets of target genes controlled by antioxidant response elements (AREs). The versatile biological functions of the different Nrf1 isoforms rely on their structural domains, harbouring conserved motifs. Notably, Nrf1 contains two transactivation domains: AD1 (acidic domain 1) and AD2 (acidic domain 2). AD1 comprises amino acids 125–298 and functions as the major transactivation domain. AD2 (amino acids 403–455) further enhances Nrf1’s transactivation activity and is of particular significance for the truncated 65 kDa isoform. Moreover, Nrf1 encompasses the Nrf2–ECH homology 1-like region, spanning amino acids 581–730. Within this region, both CNC (Cap‘n’Collar) and bZIP (Basic-region leucine zipper) domains collectively function as the DBD (DNA-binding domain). The different Nrf1 isoforms play crucial roles in maintaining redox and proteasome homeostasis, protecting against oxidative stress and regulating hepatic fatty and amino acid metabolism [26,27]. In line with this, the deglycosylated and cleaved 85 kDa form of Nrf1 serves as the primary transcriptional activator, while the shorter fragments ranging from 36 to 25 kDa act as repressors [28]. In addition to its role in redox homeostasis, Nrf1 is implicated in maintaining lipid and cholesterol homeostasis by acting as a cholesterol sensor, particularly in the liver. Here it binds to cholesterol via its N-terminal CRAC (cholesterol-recognition/amino acid consensus motif) domain, thereby modulating hepatic inflammatory signalling, adaptive metabolic responses and promoting cholesterol removal. This indicates that Nrf1 plays a crucial role in maintaining hepatic cholesterol homeostasis and protecting the liver from excessive cholesterol accumulation [29].

Here, we investigate the HCV-Nrf1 crosstalk. We study the impact of HCV on Nrf1 expression and how modulation of Nrf1 expression interferes with HCV morphogenesis. Therefore, we take a close look at the viral replication and production of progeny virions upon Nrf1 overexpression and silenced Nrf1 expression. We found out that Nrf1 overexpression restricts multiple steps of the HCV life cycle. This effect can be reversed upon Nrf1 silencing. Moreover, our study revealed that the 85 kDa Nrf1 fragment can be withdrawn from the nucleus by association with the sMaf proteins. Therefore, the overall function of Nrf1 is weakened, which results in impaired activation of Nrf1/ARE-dependent genes as well as intracellular cholesterol accumulation, favouring HCV morphogenesis.

Taken together, our data indicate strong crosstalk between HCV and Nrf1 affecting the HCV life cycle and HCV-associated pathogenesis. In light of this, Nrf1 could serve as a target for interference with the viral morphogenesis.

## 2. Materials and Methods

### 2.1. Cell Culture

The Huh7-derived cell line Huh7.5 (RRID:CVCL_7927) [30] was used for electroporation in all experiments due to its susceptibility to HCV infection and replication. The HCV genomic RNA was generated by in vitro run-off transcription of linearised plasmid DNA (pFK-pJFH1/GND and pFK-JFH1/J6) using T7 Scribe Standard RNA IVT Kit (Biozym, Hessisch Oldendorf. Germany) according to the manufacturer’s protocol. Huh7.5 cells’ electroporation with HCV RNAs was performed as described in [31]. Cells were cultivated in Dulbecco’s modified Eagle’s medium (complete DMEM) supplemented with 2 mM L-glutamine (Bio&Sell, Feucht, Germany, K0283), 100 μg/mL streptomycin, 100 U/mL penicillin and 10% *v*/*v* foetal bovine serum; (FBS.S 0615, Bio & Sell GmbH, Feucht, Germany) at 37 °C with 95% relative humidity and 5% CO_2_. HCV-replicating and non-replicating cells were passaged two to three times a week, no longer than passage six.

### 2.2. Plasmids

Plasmids encoding HCV DNA: pFK-pJFH1/GND (replication deficient) and pFK-JFH1/J6 have been previously described [31,32]. pFK-Luc-Jc1 was kindly provided by R. Bartenschlager, Heidelberg, Germany, and encodes a bicistronic reporter construct of the full-length Jc1 genome [32]. Plasmid pGreenFire1-LXRE in LXR alpha was ordered from IBA Lifesciences. Luciferase reporter construct pNQO1luc harbouring the AREs from NAD(P)H-dependent quinone oxidoreductase 1 (pNQO1luc) was used as described in [33]. Plasmid encoding EGFP (GenBank: AAB02572.1) was purchased from Clontech Laboratories Inc., Mountain View, CA, USA. The Δa1pEGFP-N1 (pJo23) vector, which is based on the pEGFP-N1 plasmid (GenBank: U55762.1) lacking the start codon, was purchased from Invitrogen, Carlsbad, CA, USA. Plasmids encoding 85 kDa or 25 kDa Nrf1 tagged with a C-terminal EGFP were constructed by inserting human Nrf1 DNA into the Δa1pEGFP-N1 vector. All the Nrf1 fragments were amplified via polymerase chain reaction using the Q5^®^ High-Fidelity DNA Polymerase (NEB #0491S), and the plasmid DNA Flag-Nrf1-HA (#34708; Addgene) was used as a template.

The following primers were used: 85 kDa-fwd (5′-AAAAAGCTTATGGTTCACCGAGACCCAGAGG-3′), 25 kDa-fwd (5′-AAAAAGCTTATGAGGCACCCAGTGCCCTG-3′) and rev (5′-AAAGGTACCCTTTCTCCGGTCCTTTGGCTTC-3′). Inserts were cloned into the backbone linearized with HindIII (NEB #R0104L) and KpnI (NEB #R0142M) restriction enzymes according to Gibson et al. [34]. Plasmids encoding human sMafG protein (Gene ID: 4097) with nuclear export/localisation signal N-terminally tagged with mCherry (GenBank: AAV52164) (pEM_Z55_mCherry_sMaf-NES; pcDNA3.1(-) sMafG-NLS) were previously generated in our lab.

### 2.3. Transient Transfection

Huh7.5, GND and Jc1 electroporated cells were transfected using linear polyethyleneimine (PEI) (1 mg/mL) (23966-100; Polysciences, Warrington, PA, USA) or FuGENE^®^ HD Transfection Reagent (E2311; Promega, Madison, WI, USA). The cells were seeded one day before the transfection at a density of 3 × 10^5^ cells/well in 6-well plates. For CLSM analyses, the cells were seeded on coverslips (1.5 H) (Carl Roth, Karlsruhe, Germany) in 12-well plates at a density of 1.5 × 10^5^ cells/well. A DNA:PEI mass ratio of 1:6 was used, followed by an 8 h incubation time and medium change. A DNA:FuGENE mass ratio of 1:3 was used, followed by a 24 h incubation and medium change. Cells were harvested 48 h after transfection, at the peak point of Nrf1 overexpression.

Nrf1 knockdown experiments were performed using the siPORT^TM^ NeoFX^TM^ Transfection Agent (Invitrogen, Carlsbad, CA, USA) according to the manufacturer’s protocol using 2 nM Nfe2l1 siRNA (SMARTPool M-019733-01-0010; Dharmacon, Lafayette, CO, USA) or scrambled siRNA (sc-37007, Santa Cruz Biotechnology, Dallas, TX, USA) as a control. The transfections were performed in 12-well plates using the overlay protocol. For one well, a Transfection mix of 0.2 µL Nfe2l1 siRNA (10 µM) or scrambled siRNA (10 µM) was prepared in 49.8 µL Opti-MEM^®^ I (Thermo Fisher, Waltham, MA, USA) and 3 µL siPORT^TM^ NeoFX^TM^ Transfection Agent were mixed with 47 µL Opti-MEM^®^ I (31985062, Thermo Fisher, Waltham, MA, USA). After combining the solutions, RNA was prepared in OptiMEM^®^ medium (31985062; Thermo Fisher, Waltham, MA, USA) using the overlay protocol siPORT (AM4510; Thermo Fisher, Waltham, MA, USA) and incubated for 15 min at room temperature. The RNA/siPORT^TM^ NeoFX^TM^ Transfection Agent complexes were evenly dispersed in 12-well plates. A total of 100 µL per 12 wells of transfection mix was pipetted into the wells and dispersed evenly. Afterwards, 900 µL of 1 × 105 stably HCV-replicating Huh7.5 cells were laid over the transfection mix. Samples were analysed 96 h after the transfection via Western blot or the CLSM method, and silencing of the Nrf1 protein was confirmed using an antibody binding to the N-terminus of the protein (#8052, Cell Signaling, Danvers, MA, USA).

### 2.4. RT-qPCR

Total RNA was isolated using RNA-solv^®^ Reagent (R6830-02; VWR) according to the manufacturer’s protocol. Equal amounts of total RNA (4 µg) were treated with RQ1 RNase-Free Dnase (M6101; WVR) and transcribed into cDNA using a random hexamer primer and RevertAid H Minus reverse transcriptase (EP0451; Thermo Fisher Scientific, Waltham, MA, USA). qPCR of 10-fold diluted samples was performed using the SYBR-green detection kit (K0251; Thermo Fisher Scientific, Waltham, MA, USA) and specific primers. Extracellular viral RNA was isolated from the cell culture supernatants via the QIAmp® Viral RNA Mini Kit (Qiagen, Hilden, Germany) and detected using LightMix Modular Hepatitis C Virus Kit (53-0557-96, TIB MOLBIOL, Berlin, Germany) in combination with LightCycler Multiplex RNA Virus Master (6754155001; Roche Diagnostics) according to the manufacturer’s protocol. All RT-qPCR experiments were performed using the LightCycler 480 Instrument II (Roche Diagnostics) and analysed using the LightCycler480 Software (v1.5.1; Roche Diagnostics).

### 2.5. Virus Titration

Virus titres were analysed based on limited dilution by determining the half-maximal tissue culture infectious dose (TCID_50_) as described previously [35]. Briefly, Huh7.5 cells were seeded in a 96-well plate at a density of 1 × 10^4^ cells/well. For extracellular TCID_50_, cells were infected using a serial dilution of cell culture supernatant (5 steps, 1:5 ratio) in 6 replicates. For intracellular TCID_50_, cells were washed with PBS, trypsinised and pelleted. Pellets were then resuspended in 1 mL DMEM and subjected to 4–5 freeze/thaw cycles at −80 °C and 37 °C, respectively. Afterwards, Huh7.5 cells were infected with the obtained supernatant (5 steps, 1:5 ratio) in 6 replicates. Cells were cultivated for 72 h. Fixation was performed with 4% formaldehyde in PBS at RT. Cells were then washed with PBS and incubated overnight at 4 °C with the NS5A-specific antiserum to detect HCV-positive cells. Horseradish peroxidase–coupled donkey-α-rabbit IgG (NA934; GE Healthcare, Chicago, IL, USA) was used as a secondary antibody, and subsequent staining was performed using 3-amino-9-ethylcarbazol (30 mM Na-acetate, 12 mM acetic acid, 0.05% *w*/*v* 3-amino-9-ethylcarbazol, 0.01% H_2_O_2_) [36]. The resulting TCID_50_ was calculated based on the method of Spearman and Kärber [37,38].

### 2.6. Luciferase Reporter Gene Assay

Huh7.5 and HCV-replicating Huh7.5 cells were transfected using pGreenFire1-LXRE plasmid (encoding for an LXR promotor-driven firefly luciferase) or pNQO1luc (encoding for a NQO1 promotor-driven firefly luciferase). Then, 48 h after transfection, cells were lysed with 200 μL of luciferase lysis buffer (25 mM Tris-HCl, 0.1% Triton-X 100, 2 mM DTT, 2 mM EGTA, 10% glycerol, pH 7.5). The luciferase activity of lysates (50 µL) was analysed by the addition of luciferase substrate (25 mM Tris-HCl pH 7.8, 5 mM MgCl_2_, 33.3 mM DTT, 0.1 mM EDTA, 470 μM Luciferin, 530 μM ATP) and subsequent detection in the Orion II microplate Luminometer (Titerek Berthold, Pforzheim, Germany). The samples were analysed as technical duplicates. Luciferase levels were referred to the protein concentration of the appropriate lysates, determined by Bradford assay [36]. To monitor HCV replication, Huh7.5 cells were electroporated with the bicistronic replicon pFK-Luc-Jc1, and a luciferase assay was performed as described previously [39].

### 2.7. Immunofluorescence Analysis

Huh7.5 and HCV-replicating Huh7.5 cells were grown in 12-well plates on coverslips (1.5H) (Carl Roth, Karlsruhe, Germany). Thereafter, cells were washed with PBS and fixed with 4% formaldehyde for 20 min at RT. These were then blocked for 15 min with 5% bovine serum (Carl Roth, Karlsruhe, Germany, T844.4) in TBS-T and incubated for 1h with primary and secondary antibodies in a humidity chamber, respectively. As a diluent, a blocking solution was used. After washing with TBS-T, coverslips were mounted on glass slides with Mowiol 40-88 (324590-100G; Sigma Aldrich, St. Louis, MO, USA). Nuclei were stained with DAPI (4.6-diamidino-2-phenylindole) (6335.1; Roth) solution (1 μg/μL). For lipid droplet visualisation, Nile Red (Cay30787-250; Biomol) solution (1 μM) was added during the secondary antibodies incubation. F-Actin was stained with phalloidin-Atto 633 (68825, Sigma Aldrich, St. Louis, MO, USA, 1:500). Filipin III (F4767-1MG; Sigma Aldrich, St. Louis, MO, USA) was used to stain total intracellular cholesterol. First, cells were fixed as mentioned above. Afterwards, Schiff-base from FA-fixation was quenched with TBS for 5 min at RT. Cells were then permeabilized with 0.05% TritonX-100 (T9284; Sigma Aldrich, St. Louis, MO, USA) and blocked in 5% bovine serum in TBS containing 0.05 mg/mL filipin. These were then incubated with the primary and secondary antibodies with 0.05 mg/mL filipin in a humidity chamber. After washing with TBS, coverslips were mounted as described above. Immunofluorescence staining was analysed using a confocal laser scanning microscope (Leica TCS SP8 System with a DMi8 microscope) and Las X Control Software (Leica, Wetzlar, Germany). The used objectives were 100×, numerical aperture 1.46. Total fluorescence per cell was calculated using ImageJ software [40] and the following formula: corrected total cell fluorescence (CTCF) = integrated density—(area of selected cell × mean fluorescence of background readings). In total, a minimum of ten cells were measured.

### 2.8. Antibodies

Primary antibodies were raised against HCV core (MA1-080; Thermo Scientific, Waltham, MA, USA), NS3 (VIS-1847-100; Biozol/MyBioSource), β-actin (A5316; Sigma-Aldrich), C-terminal Nrf1 (PA590023; Thermo Scientific, Waltham, MA, USA and sc-28379; Santa Cruz), N-terminal Nrf1 (#8052; Cell Signaling, Danvers, MA, USA) and sMaf G/F/K (sc-22831; Santa Cruz). For NS5A detection, polyclonal rabbit-derived serum was used.

Secondary antibodies directed against mouse or rabbit were produced in donkey and were conjugated to Alexa Fluor 488 (28115; Invitrogen, Carlsbad, CA, USA), Alexa Fluor 546 (A10036; Invitrogen, Carlsbad, CA, USA or A10040; Thermo Fischer Scientific) and Alexa Fluor 633 (A-21052; Thermo Fischer Scientific, Waltham, MA, USA).

### 2.9. Immunohistochemistry

An immunostaining procedure was performed on paraffin-embedded human liver sections, obtained from patients chronically infected with HCV genotype 1a and healthy individuals as controls. Both infected and healthy individuals were male, aged between 45 and 58. The specimens were kindly provided by K. Klingel, Institute of Molecular Pathology, Universitätsklinikum Tübingen, Tübingen, Germany. Sample collection was supervised by the Ethics Committee at the Medical Faculty of the Eberhard Karls University and at the University Hospital of Tübingen. Samples were fixed with 4% formaldehyde in PBS. Then, 4 µm thick paraffin liver sections were deparaffinised for 15 min with xylene, 10 min with 99% ethanol, 10 min with 75% ethanol and 5 min in ddH_2_O. Antigen retrieval was performed by heating the sections in 10mM sodium citrate buffer, pH 6.0, at 95 °C for 30 min. These were then blocked for 60 min in 10% BSA with 0.1% Tween20. The primary antibodies used for staining were the anti-Nrf1 antibody, detecting the N-terminus of the protein (#8052; Cell Signalling, Danvers, MA, USA), and the anti-core antibody (MA1-080; Thermo Scientific, Waltham, MA, USA), detecting HCV-positive cells. Anti-rabbit Alexa Fluor 488-conjugated (28115; Invitrogen, Carlsbad, CA, USA) and anti-mouse Alexa Fluor 546-conjugated (A10036; Invitrogen, Carlsbad, CA, USA) were used as secondary antibodies. Nuclei were stained with DAPI (4.6-diamidino-2-phenylindole) (6335.1; Roth) solution (1 μg/μL). All the dilutions were performed in the blocking solution. After washing with TBS-T, liver sections were mounted on glass slides with Mowiol 40-88 (324590-100G; Sigma Aldrich, St. Louis, MO, USA). Staining was analysed using a confocal laser scanning microscope (Leica TCS SP8 System with a DMi8 microscope) and Las X Control Software (Leica, Wetzlar, Germany).

### 2.10. SDS-PAGE and Western Blot Analysis

Cells were lysed using RIPA buffer (50 mM Tris-HCl pH 7.2, 150 mM NaCl, 0.1% SDS (*w*/*v*), 1% sodium deoxycholate (*w*/*v*) and 1% Triton X-100) containing protease inhibitors, followed by sonication. An equal amount of protein was denatured in sample 1x SDS loading buffer, boiled for 10 min at 95 °C and separated by SDS-PAGE electrophoresis with 10% *v*/*v* acrylamide. Afterwards, proteins were transferred onto a polyvinylidene difluoride (PVDF) membrane (P667.1, Roth) and blocked in 1× ROTI-Block solution (A151.2, Roth) [36]. Detection of the chosen proteins was conducted using the following primary antibodies: anti-GFP (632592; Takara, San Jose, CA, USA), anti-mCherry (ab167453; Abcam, Cambridge, UK), anti-αTubulin (sc-5546; Santa Cruz) anti-NS3, anti-NS5A, anti-Nrf1 and anti-βactin as described above. Secondary antibodies directed against mouse or rabbit were coupled to IRDye680RD or IRDye800CW (926-68073, 926-32213; LI-COR Biosciences, San Jose, CA, USA) and horseradish peroxidase (NA934, NXA931; GE Healthcare). All the antibody dilutions were performed in the blocking reagent and incubated for 1h on the membrane. After washing with TBS-T, fluorescence was detected by the LI-COR Odyssey CLx scanner (LI-COR Biosciences, San Jose, CA, USA). After incubation with Luminata Forte Western HRP substrate (WBLUF0100; Merck Chemicals GmbH Millipore GmbH, Darmstadt, Germany), chemiluminescence was detected by the INTAS-Imaging System (Intas Pharmaceuticals Limited, Ahmedabad, India). Data was analysed using Image-Studio Lite v5.2.5 (LI-COR Biosciences, San Jose, CA, USA).

### 2.11. Determination of Half-Life

To inhibit protein synthesis, HCV-replicating and -negative cells were treated with 142 μM cycloheximide (C7698-5G; Sigma Aldrich, St. Louis, MO, USA). Cells were lysed using RIPA buffer (50 mM Tris-HCl pH 7.2, 150 mM NaCl, 0.1% SDS (*w*/*v*), 1% sodium deoxycholate (*w*/*v*), 1% Triton X-100) containing protease inhibitors, at different time points between 0 and 4 h after cycloheximide treatment, and analysed by Western blot. Western blot analysis of cellular lysates derived from HCV-positive and HCV-negative cells was performed using an antibody binding to the N-terminus of the protein (#8052, Cell Signaling, Danvers, MA, USA), detecting 140-120 kDa Nrf1, representing full-length protein. The protein amount of Nrf1 was normalised to beta-actin, and the half-life of Nrf1 was determined by exponential regression.

### 2.12. Kinome Analysis

Huh7.5 cells stably electroporated with HCV RNAs were seeded in a 6-well plate with a density of 3 × 10^5^ cells/well and cultivated in fully supplemented DMEM medium. After cells adhered, transfection of plasmid DNA encoding Nrf1 constructs was carried out using FuGENE^®^ HD Transfection Reagent as described earlier. Medium exchange was performed 24 h post-transfection and cells were lysed 72 h post-transfection using M-PER™ Mammalian Protein Extraction Reagent (78503, Thermo Scientific, Waltham, MA, USA) supplemented with Halt™ Protease Inhibitor Cocktail (87785, Thermo Scientific, Waltham, MA, USA) and Halt™ Phosphatase Inhibitor Cocktail (78420, Thermo Scientific, Waltham, MA, USA). Similar protein amounts, as assessed via the Pierce™ BCA Protein Assay Kit (23225, Thermo Scientific, Waltham, MA, USA), of cleared lysates were then used to determine relative changes in peptide phosphorylation and subsequent analyses of upstream kinases. Measurements were carried out on peptide arrays comprising distinct sets of peptides being targets of certain kinases using PamChips^®^ and respective reagent kits (32516, 32112, 32501, 32201, PamGene International, Hertogenbosch, The Netherlands) in combination with the PamStation^®^12 instrument operated with Evolve2 software. Active kinases in lysates phosphorylate the immobilised peptides, which were visualised via fluorescently conjugated antibodies and a CCD camera. Relative changes in peptide phosphorylation, along with statistics, were then computed using the BioNavigator v6.3.67 software. These were used to predict the activity of upstream kinases, making use of distinct peptides as substrates, with the help of public databases (PhosphoNet, published in vitro or in vivo experiments or Kinexus), as described elsewhere [41]. Changes in peptide phosphorylation were considered significant below a *p*-value of 0.05, whereas upstream kinases were considered relevant above a final score of 2 (combinatory score of kinase specificity, the extent of relative change in activity and the corresponding statistics). Finally, significantly deregulated kinases were screened for their presence in certain gene ontology terms (GO terms) retrieved from the EMBL-EBI database as of 4 December 2022 (http://purl.obolibrary.org/obo/go/releases/2022-12-04/go.owl). GO terms included covered inflammatory response (GO:0006954), innate immune response (GO:0045087), cholesterol biosynthetic process (GO:0006695) and lipid biosynthetic process (GO:0008610).

### 2.13. Lipid Droplet Analysis

LD analysis was performed using the particle analysis feature in Fiji (Image J) open-source analysis software [40]. The size of the particle was set as 0.01-infinity (inch^2). Circularity was set as 0.00–1.00. The total count and perimeter were measured. The total volume of lipid droplets was calculated based on the average number of LDs multiplied by the average volume of LDs. Additional equations used in the calculations include$$V=\frac{4}{3}\pi r^{3}$$$$r=\frac{1}{2}d$$d=πP
where *V* is the volume, *r* is the radius, *d* is the diameter, and *P* is the perimeter.

### 2.14. Statistical Analysis

Similar conditions were applied for all experiments. Each figure legend shows the number of independent experiments that correspond to that figure. Prism v9.2 software (GraphPad Prism version 9.2 for Windows, GraphPad Software, Boston, Massachusetts, USA, www.graphpad.com) was used to perform statistical analysis and plot the graphical representation of the data. The results are described as the mean ± standard error of the mean (SEM). For statistical comparisons, a normality test was performed using the Shapiro–Wilk conditions. For normally distributed data, statistical significance was calculated by the unpaired *t*-test. Data not showing a Gaussian distribution were tested using the Mann–Whitney test. Statistical significance is displayed as stated: ns—not significant, * *p* < 0.05, ** *p* < 0.01, *** *p* < 0.001, **** *p* < 0.0001. The threshold for the *p*-value was set using the Holm–Šídák method.

## 3. Results

### 3.1. Decreased Amount of Full-Length Nrf1 in HCV-Replicating Cells

Infection with HCV leads to elevated levels of ROS and disturbed lipid metabolism [39,42,43,44]. Our group’s previous work revealed that HCV impairs Nrf2/ARE-dependent gene expression. Here, HCV core mediates the translocation of sMaf proteins from the nucleus to the replicase complex on the cytoplasmic face of the ER. This prevents the entry of Nrf2 into the nucleus and thereby inhibits the Nrf2-dependent expression of cytoprotective genes [33].

To investigate the impact of HCV on the expression of the NRF1 gene, we determined the number of Nrf1-specific transcripts in HCV-replicating cells (Jc1) and the corresponding HCV-negative control cells (GND) by qPCR. Nrf1 expression, reflected by the level of mRNA, was significantly induced in the Jc1 cells (~2-fold) compared to control cells (Figure 1a). To further study the impact of HCV on the protein level of Nrf1, we performed Western blot analysis of cellular lysates derived from HCV-positive and HCV-negative cells. In contrast to the observed increase in Nrf1-specific transcripts, the protein amount of full-length Nrf1 was significantly reduced in HCV-positive cells (~0.4-fold) (Figure 1b).

To further corroborate the results of the Western blot analysis and investigate the impact of HCV on the intracellular distribution of Nrf1 protein in HCV-positive cells, we performed CLSM analysis. In accordance with the Western blot analysis, the immunofluorescence data confirmed decreased amounts of endogenous Nrf1 in HCV-positive cells compared to the control (GND) (Figure 1c and Appendix A). Interestingly, an Nrf1-specific antibody that specifically binds to the C-terminal part of the protein, and therefore is able to detect the different cleaved Nrf1 isoforms, indicates that less cleaved Nrf1 can be found inside the nucleus of the Jc1 cells compared to the GND cells (Figure 1d). Nuclear localisation is specific for cleaved Nrf1 isoforms, as they lack the N-terminal transmembrane domain and are no longer anchored in the ER [22].

Moreover, in vivo data corroborate these observations. Namely, CLSM analysis of liver sections derived from a patient with chronic HCV infection revealed a significantly lower Nrf1 protein level in the patient suffering from chronic HCV compared to the healthy patient (Figure 1e).

Due to the discrepancy between the increased number of Nrf1-specific transcripts and the decreased Nrf1 protein amount, we investigated whether the stability of Nrf1 in HCV-positive cells decreased. For this purpose, we determined the half-life of the full-length Nrf1 in Jc1 and GND cells by incubation with 142 µM CHX to block translation. The amount of the 120–140 kDa Nrf1 isoforms was detected after different time points by Western blot using an antibody binding to the N-terminus of the protein. Surprisingly, the data reveal no changes in the half-life of Nrf1 in the Jc1 cells (~45 min) compared to the GND cells (~43 min) (Figure 1f). Based on this, we speculate that an impairment in Nrf1 translation is causative for the decreased Nrf1 protein amount in HCV-positive cells.

To sum up, the above data indicate that HCV infection leads to increased levels of Nrf1-specific transcripts and a decreased amount of the full-length Nrf1 protein.

### 3.2. Silencing of Nrf1 Expression Favours HCV Replication

To further characterise the impact of Nrf1 on the HCV life cycle, the expression of Nrf1 was silenced by transfection of HCV-positive cells with an Nrf1-specific siRNA. Transfection with scrambled RNA (scrRNA) served as a control. Efficient silencing of already diminished levels of Nrf1 in HCV-positive cells was confirmed by Western blot analysis (Figure 2a). As the limited transfection efficiency of the siRNA affects the analysis methods based on the total cell lysates, single-cell analyses were performed based on immunofluorescence staining and quantitative CLSM. The proviral effect of the silencing of Nrf1 was further corroborated by quantitative single-cell analysis using CLSM. Here, a significant increase in the amount of HCV core (Figure 2b) and NS3 protein levels was observed (Figure 2c). To study the impact on the formation of infectious viral particles, the intra- and extracellular levels of infectious viral particles were quantified by determining the TCID_50_ (Figure 2d,e). This revealed a significant increase in the number of infectious intracellular viral particles and a slight increase in released viral particles if Nrf1 expression was silenced. Furthermore, silencing of Nrf1 expression significantly enhanced viral replication as evidenced by a luciferase reporter virus (Figure 2f).

Taken together, these data indicate that silencing of Nrf1 expression favours HCV replication and subsequent progeny virion formation.

### 3.3. Overexpression of Specific Nrf1 Fragments Restricts HCV Replication, Assembly, and Release

To further study the relevance of Nrf1 for the HCV life cycle, the 85 kDa fragment of Nrf1 was overexpressed in HCV-positive cells. Among the different Nrf1 isoforms, the 85 kDa isoform is considered the paradigm for the active form of the protein. The 85 kDa Nrf1 encompasses the transcriptional activator and the DNA binding domain (DBD), thus functioning as a main activator for the transcription of the Nrf1 target genes [28]. CLSM-based single-cell analysis via the calculation of the CTCF indicates a significant reduction in the amount of NS5A (~0.6-fold) and HCV core (~0.6-fold) in the cells overexpressing the 85 kDa fragment of Nrf1 (Figure 3a,b).

To further characterise the impact of Nrf1-85 kDa overexpression on HCV assembly and release, the number of intra- and extracellular viral genomes and infectious viral particles was studied. Overexpression of the 85 kDa Nrf1 fragment leads to a significant increase in the intracellular viral genomes (~2-fold) and a significant decrease in the amount of extracellular viral genomes (~0.3-fold) (Figure 3c,d) compared to the Mock control. Interestingly, 85 kDa Nrf1 overexpression significantly decreased the amount of intra- and extracellular infectious viral particles (Figure 3e,f) resulting in a decreased specific infectivity, described by the ratio of infectious viral particles (TCID_50_/_mL_) to viral genomes (genomes/mL) (Figure 3g). Moreover, overexpression of the 85 kDa Nrf1 significantly decreased viral replication (~0.3-fold) as evidenced by a luciferase reporter virus (Figure 3h).

In summary, these data indicate that overexpression of the 85 kDa Nrf1-specific fragments impairs the formation of intact viral particles and their release. This could be causative for the elevated amounts of intracellular viral genomes, which, due to the lack of HCV core and impaired viral assembly, are not accessible for the formation of mature viral particles.

### 3.4. Extracellular sMaf Proteins Have the Capacity to Withdraw Nrf1 from the Nucleus

In HCV-replicating cells, sMaf proteins are delocalized from the nucleus to the replicase complex by binding to NS3 on the cytoplasmic face of the ER [33]. The withdrawal of sMaf proteins to the replicase complex prevents Nrf2, an important factor in redox homeostasis, from entering the nucleus. Nrf2 is trapped at the replicase complex, which leads to an impaired expression of Nrf2/ARE-dependent genes in HCV-replicating cells. It has been established that, along with Nrf2, Nrf1 protein also dimerises with sMafs via the bZIP domain [22]. Based on this, we investigated whether the observed delocalisation of sMaf proteins in HCV-positive cells likewise affects the localisation of Nrf1. To answer this, we transfected HCV-positive and -negative cells with a plasmid encoding 85 kDa Nrf1 fused to a C-terminal EGFP. CLSM analyses revealed nuclear localisation of 85 kDa Nrf1 in both HCV-positive and -negative cells. Interestingly, in the cells overexpressing the 85 kDa Nrf1, we found that the sMaf proteins outside the nucleus were independent of HCV (Figure 4a indicated by green arrow), whereas in the absence of 85 kDa Nrf1 overexpression, sMafs were localised in the nucleus (Figure 4a indicated by blue arrow). This was not the case in the cells that were transfected with the control plasmid. Here, the sMafs were completely localised in the nucleus (Appendix A).

The observed nuclear localisation of Nrf1 in HCV-positive cells raised the question whether, in contrast to Nrf2, the 85 kDa Nrf1 fragments cannot be withdrawn from the nucleus by sMaf proteins. Another cause of the observed effect might be the fact that, due to the strong overexpression of 85 kDa Nrf1, extracellular sMafs are already saturated. To investigate this, expression vectors encoding for fusion proteins of mCherry-sMaf and a nuclear export signal (sMaf-NES) or a nuclear localisation signal (sMaf-NLS) were generated. Co-transfection of the mCherry-sMaf-NES construct together with the plasmid encoding for the 85 kDa Nrf1-EGFP fusion protein revealed that Nrf1 can be efficiently translocated from the nucleus to the cytoplasm by the sMaf-NES proteins in HCV-positive and HCV-negative cells (Figure 4b).

Vice versa, co-transfection of the 85 kDa Nrf1-EGFP with the mCherry-sMaf-NLS construct resulted in a complete translocation of Nrf1 to the nucleus in HCV-positive and HCV-negative cells (Figure 4c).

These data indicate that 85 kDa Nrf1 has the capacity to interact with sMaf proteins and thus can be withdrawn from the nucleus by delocalized sMaf proteins in HCV-positive and HCV-negative cells.

### 3.5. Overexpression of sMaf-NES Leads to an Impaired Activation of Nrf1/ARE-Dependent Gene Expression by Extranuclear aMaf Variant

In the next step, the functional implications of the delocalisation of sMaf proteins and their decreased amount in HCV-positive cells on the Nrf1-dependent expression of ARE-driven genes were studied. Therefore, reporter gene assays were performed. For this purpose, a luciferase reporter gene under the control of the ARE sites of the NAD(P)H Quinone oxidoreductase 1 promoter was used (pLucNQO1). The reporter gene assay confirmed the decreased activation of ARE-dependent gene expression in HCV-positive cells (Jc1) compared to the HCV-negative control (GND) (Figure 5). This is in accordance with our previous reports describing reduced expression of ARE-dependent genes in HCV-positive cells [19,21,45].

Overexpression of the 85 kDa Nrf1 caused a strong increase in the reporter gene expression in HCV-negative cells (Figure 5). In contrast to this, in HCV-positive cells, overexpression of the Nrf1-85 kDa form only caused a smaller increase in the reporter gene expression (Figure 5). This might be due to the lack of nuclear sMaf in HCV-positive cells.

To investigate whether Nrf1/ARE-dependent NQO1 induction is modulated by the presence of nuclear sMaf proteins, we further co-transfected HCV-positive and the corresponding HCV-negative cells with the reporter construct, 85 kDa Nrf1, and sMaf-NES/NLS constructs. Overexpression of sMaf-NES resulted in a significant decrease in NQO1 expression in the Mock-transfected control. However, overexpression of sMaf-NES and 85 kDa Nrf1 had no great impact on NQO1 expression compared to the cells expressing endogenous sMaf levels. Yet, in all settings, the induction of the reporter gene was impaired in the Jc1 cells, reflecting the persistent lack of nuclear sMaf proteins in HCV-positive cells (Figure 5).

Vice versa, overexpression of sMaf-NLS was able to rescue the expression of NQO1 in the Mock-transfected HCV-positive cells. Moreover, after co-expression of sMaf-NLS and 85 kDa Nrf1, we observed a significant increase in NQO1 activation in both HCV-positive and -negative cells (Figure 5).

Based on these observations, we conclude that withdrawal of sMaf proteins from the nucleus impairs Nrf1/ARE-dependent gene expression. Moreover, sMaf-NES proteins represent a promising and specific tool to experimentally modulate Nrf1-dependent gene expression.

### 3.6. Impaired LXR Activation in HCV-Positive Cells

The HCV life cycle is tightly connected to the host cell cholesterol/lipid metabolism [46]. In general, a variety of mechanisms differentially control intracellular cholesterol/lipid levels. Thus, intracellular cholesterol accumulation is detected by a variety of sensors within the endoplasmic reticulum, triggering a response in order to induce the export of excess cholesterol. One of these factors involved in the cholesterol removal programme is the liver X receptor alpha (LXR-α). To investigate LXR activation in HCV-positive and HCV-negative cells, the cells were transfected with a reporter gene construct harbouring the luciferase gene under the control of the LXR-α promoter. The activity of the LXR promoter was significantly decreased in HCV-positive compared to HCV-negative cells (Figure 6a).

According to a model formulated by Widenmaier et al. [28], Nrf1 is likely to be involved in the regulation of LXR activity. Therefore, we investigated whether overexpressing the transcriptionally active Nrf1 form (the 85 kDa fragment) would rescue LXR promoter activity in HCV-positive cells. However, we found that overexpression of the 85 kDa Nrf1 fragment only slightly increased LXR-α promoter activation in both HCV-positive and HCV-negative cells. Consequently, the direct relevance of Nrf1 to the HCV-dependent inhibition of the LXR promoter remains unclear (Figure 6a).

### 3.7. Elevated Cholesterol Levels in HCV-Positive Cells

As LXR is a relevant factor for cholesterol removal from the cells, we investigated whether the decreased activation of LXR expression in HCV-positive cells is reflected by an enhanced intracellular cholesterol level. To analyse the intracellular cholesterol level, we stained with filipin III, which specifically binds to unestrified cholesterol forming a fluorescent complex. Quantitative CLSM analysis of the filipin-specific fluorescence revealed significantly elevated intracellular cholesterol levels in HCV-positive Jc1 cells compared to the GND control (Figure 6b). As described above, LxR activity could be regulated by Nrf1. Therefore, to further characterise the impact of impaired Nrf1 functionality on the intracellular cholesterol level, we transfected the cells with a plasmid encoding for a 25 kDa isoform of Nrf1 (25 kDa Nrf1-EGFP). The 25 kDa Nrf1 isoform acts as a dominant-negative mutant and therefore impairs Nrf1 functionality [28]. Indeed, overexpression of 25 kDa Nrf1 caused a significant increase in the intracellular cholesterol in HCV-negative cells compared to the Mock control (Figure 6c). In HCV-positive cells, only a weak increase in the already notably higher cholesterol level was observed (Figure 6c). This might reflect the already impaired Nrf1 activity in HCV-positive cells.

Taken together, these data suggest that impaired Nrf1 functionality contributes to an elevated intracellular cholesterol level, observed in HCV-positive cells. The impairment in Nrf1 functionality could be a factor that contributes to reduced LXR activation and expression in HCV-positive cells. This in turn could be a mechanism that, by defective cholesterol export, leads to elevated intracellular cholesterol levels observed in HCV-positive cells or HCV-negative cells with impaired Nrf1 functionality. The connection, however, cannot be clearly stated and requires follow-up experiments.

### 3.8. Modulation of Nrf1 Activity Directly Affects the Number of Lipid Droplets

To further investigate the crosstalk between HCV and Nrf1 and its impact on the host’s lipid metabolism, lipid droplets (LDs) were studied next. LDs play a crucial role in later steps of the HCV life cycle, serving as a platform for viral assembly, thereafter allowing the progeny virions to be released [47]. In light of this, the size and the number of LDs in HCV-positive and HCV-negative cells were analysed.

Interestingly, the number of LDs in HCV-positive and HCV-negative cells was found to be comparable (Figure 7a,b). Overexpression of the 85 kDa Nrf1 isoform significantly decreased the number of LDs (Figure 7b) in the HCV-positive cells compared to the Mock-transfected cells. In case of the HCV-negative cells, the effect of the 85 kDa Nrf1 overexpression was less pronounced compared to the Mock-transfected cells.

To further analyse the overall parameters of LDs, we decided to assess their total volume. The total volume of lipid droplets was calculated based on the average number of LDs multiplied by their average volume. The total volume of LDs in HCV-negative cells was slightly lower than in HCV-positive cells (Figure 7c). The volume of LDs is more strongly affected by overexpression of the 85 kDa Nrf1 form in HCV-positive compared to HCV-negative cells (Figure 7c). These results are in accordance with the more pronounced impact of overexpression of the 85 kDa Nrf1 form on the number of LDs in HCV-negative cells described above.

Taken together, these data indicate that the inhibited Nrf1 activity in HCV-positive cells has an impact on the number and volume of lipid droplets. Restoration of Nrf1 functionality in HCV-positive cells by overexpression of the 85 kDa form reduces the number and volume of LDs.

### 3.9. Inhibition of Nrf1 Modulates the Host Kinome Related to Inflammation, Innate Immunity, and Lipid Metabolism

The crosstalk between HCV and Nrf1 directly influences the activation of Nrf1-dependent gene expression. As Nrf1 is involved in the control of anti-inflammatory processes, deregulation of Nrf1 might be relevant for HCV-associated pathogenesis [29]. To further characterise impaired Nrf1 functionality, kinome analyses of HCV-positive and -negative cells, and HCV-negative cells overexpressing the dominant-negative 25 kDa-Nrf1 form, were carried out. It was investigated if there is an overlap with respect to deregulated kinases by replication of HCV or selective inhibition of Nrf1 by coexpression of 25 kDa Nrf1. Therefore, we aimed to screen for the presence of deregulation in general cholesterol/lipid metabolism and inflammation pathways. First, we identified differences in the phosphorylation status of substrate peptides in HCV-positive vs. HCV-negative cells (Figure 8a) or HCV-negative cells with selectively inhibited Nrf1 vs. non-inhibited Nrf1 (Figure 8d). Those differences were then used to pinpoint responsible kinases upstream of the substrate peptides (Figure 8b,c,e,f). Significantly affected peptides and deregulated kinases were then checked for their involvement in innate immune response, inflammatory response and, most importantly, lipid metabolism via gene ontology search.

The kinome analysis revealed a major impact on the general host kinase landscape during HCV production (Figure 8a,c) compared to HCV-negative cells (GND). Among these were numerous kinases involved in the regulation of inflammatory and innate immune processes (Figure 8b), such as Src-family kinases (SFKs) or receptor tyrosine kinases (RTKs). Interestingly, among these, the significantly deregulated was protein tyrosine kinase 6 (PTK6), which is also known to partially regulate the Akt/AMPK axis, thus controlling metabolism. To obtain an idea of whether the inhibition of Nrf1-dependent effects in HCV-positive cells is causative for the deregulation of kinases favouring inflammatory processes and affecting lipid metabolism, Nrf1 functionality was impaired in HCV-negative GND cells by overexpression of the Nrf1-25 kDa fragment. Kinome analysis of these cells was performed in comparison to Mock-transfected GND cells. While the overall effect of the overexpression of the 25 kDa fragment of Nrf1 in HCV-negative cells was less pronounced (Figure 8b,d), overlapping effects with HCV infection could be observed. Here, host defence-related kinases and those involved in lipid metabolism are deregulated (Figure 8e,f). There is an overlap between HCV replication and direct inhibition of Nrf1 by overexpression of the 25 kDa fragment in HCV-negative cells with respect to the general impact on lipid metabolism and inflammatory processes. Interestingly, there is a direct overlap with respect to the shared regulation of PTK6 and the activation of AMPK alpha subunit (PRKAA1), suggesting an impact on metabolic processes. The strong impact of the transdominant-negative 25 kDa Nrf1 fragment in GND cells on the inhibitor of the nuclear factor kappa B kinase subunit epsilon (IKBKE) reflects a potent role in the regulation of inflammatory processes.

On one hand, this shows that inhibition of Nrf1-dependent gene expression is a factor contributing to a proinflammatory kinase profile. However, on the other hand, it reveals that in addition to the inhibition of Nrf1, a variety of other factors contribute to the kinome profile of HCV-positive cells.

## 4. Discussion

The life cycle of HCV is tightly associated with lipid metabolism. HCV replication is associated with an intense rearrangement of the ER and the ‘membranous web’ formation. The ‘membranous web’ serves as the replication site and comprises ER-derived double-membrane vesicles (DMVs), multiple host factors and viral RNA and proteins [48]. HCV assembly, however, takes place in close proximity—in detergent-resistant membranes (DRMs) of the ER or mitochondrial-associated ER membranes (MAMs) [11]. A central place for HCV morphogenesis is cytosolic lipid droplets, as they serve as a platform that links the HCV replication and assembly sites [9]. The late steps of the viral life cycle are also connected with lipid metabolism, as the release pathway of HCV is described to share many steps with lipoprotein release and uptake [49]. Moreover, the mature viral particle is referred to as a lipoviroparticle due to its envelope that includes very-low-density lipoprotein (VLDL) components [50,51]. Nrf1, a transcription factor, plays a vital role in cellular homeostasis not only by responding to oxidative stress, but also as a cholesterol sensor and modulator of intracellular lipid levels [26,29]. HCV was described to modulate Nrf2/ARE-dependent gene expression by an unprecedented mechanism: in HCV-positive cells, sMaf proteins are translocated from the nucleus to the replicase complex on the cytoplasmic face of the ER by binding to NS3 protein. This traps Nrf2 at the replicase complexes and thereby prevents the entry of Nrf2 into the nucleus. Moreover, the lack of sMafs in the nucleus prevents their heterodimerysation with Nrf2 and in turn inhibits the activation of Nrf2/ARE-dependent cytoprotective genes [33]. A recent study reports that both Nrf1 and Nrf2 transcription factors co-regulate genes that protect against hepatic stress induced by excess cholesterol accumulation [52].

In light of the essential role of Nrf1 in the control of hepatic cholesterol levels and the intimate connection of the HCV life cycle with lipid metabolism, it was tempting to investigate the impact of HCV on Nrf1 expression, turnover, and localisation. Nrf1 protein amount was significantly decreased in HCV-positive cells and in liver tissues derived from a patient suffering from chronic HCV. In contrast, Nrf1 mRNA levels remained significantly increased. As we could not observe any difference in the half-life of the full-length Nrf1 protein in HCV-positive and HCV-negative cells, we assume that HCV modulates mRNA translation, as recently observed in ZIKV- or DNEV-infected cells. Singh et al. recently identified 19 or 7 repressed mRNAs in ZIKV- or DENV-infected cells, respectively. This finding is even more significant in the context of our research, as one of the translationally repressed genes in DENV-infected cells is NRF1. The regulation presumably occurs via the recruitment of various RNA-binding proteins [53].

Since the Nrf1 protein amount is decreased in HCV-positive cells, we hypothesise that the reduced levels of Nrf1 favour HCV replication. Indeed, silencing of Nrf1 was found to increase HCV replication. In contrast, overexpression of the transcriptionally active 85 kDa Nrf1 was found to decrease HCV replication. Impaired Nrf1 activity is associated with enhanced intracellular lipid levels and formation of enlarged lipid droplets. Moreover, as a cholesterol sensor, Nrf1 has recently been linked to the regulation of LxR, one of the key players in the detoxification of cells from excess cholesterol [29]. In line with this, we observe that in HCV-positive cells, the activation of the LXR-promoter is impaired. Our data suggest that impaired Nrf1 functionality contributes to an elevated intracellular cholesterol level, observed in HCV-positive cells. In principle, the impaired Nrf1 functionality in HCV-positive cells could be a factor that contributes to reduced LXR activation and expression. This in turn could be a mechanism that, by defective cholesterol export, leads to elevated intracellular cholesterol levels observed in HCV-positive cells or HCV-negative cells with impaired Nrf1 functionality. However, it cannot be excluded that the observed effects represent a more indirect crosstalk or a coincidence and do not reflect a direct correlation. Moreover, HCV replication is associated with intracellular cholesterol accumulation and an impact on lipid droplets. Rescue of Nrf1 functionality by overexpression of the 85 kDa Nrf1 reduces the number of lipid droplets in HCV-positive cells. This effect is much more pronounced in HCV-positive cells compared to HCV-negative cells. Likewise, overexpression of the 85 kDa form triggered a much more pronounced reduction in the lipid droplet volume in HCV-positive cells compared to HCV-negative cells. This might reflect the fact that in the HCV-negative cells, there is an intact Nrf1 activity regulating the number and volume of lipid droplets. In contrast to this, in HCV-positive cells, there is an impaired Nrf1 functionality. Under these conditions, the rescue of Nrf1 functionality by overexpression of the 85 kDa form exerts a much stronger effect on these parameters.

Apart from the decreased Nrf1 levels, the Nrf1/ARE-dependent gene expression is impaired in HCV-replicating cells. It is known that members of the Cap’N’Collar family, including Nrf1 and Nrf2, form heterodimers with sMaf proteins [54]. Our previous work showed that in HCV-replicating cells, due to the NS3-mediated delocalisation of sMafs to the replicase complex on the cytoplasmic face of the ER, Nrf2 was incapable of entering the nucleus [33]. Therefore, we wondered whether a similar phenomenon occurs in the case of Nrf1. Indeed, we observed significantly lower amounts of cleavage products of endogenous full-length Nrf1 in the nucleus of HCV-positive cells. However, overexpression of the GFP-tagged 85 kDa Nrf1, which is no longer associated with the ER but still has the capacity to bind to sMaf, revealed a strong nuclear localisation in HCV-positive cells. Yet, it was unclear whether the observed nuclear localisation of Nrf1 reflects its inability to bind to sMaf proteins or if the strong overexpression of Nrf1 fragments leads to the saturation of ER-localised sMaf proteins. The saturation of extranuclear sMaf with Nrf proteins (endogenous Nrf1 or Nrf2 and overexpressed Nrf1) could prevent the retention of the large excess of Nrf1-GFP fusion proteins outside the nucleus. The observation that co-expression of recombinant sMaf constructs harbouring a nuclear localisation (NLS) or a nuclear export signal (NES) clearly demonstrates the capacity of sMaf proteins to bind to Nrf1 and modulate its localisation. Moreover, these experiments show that Nrf1/ARE-dependent gene expression can be modulated by delocalised sMaf proteins. Apart from the impact on the HCV life cycle, sMaf-NES fusion proteins could represent a novel and very specific tool to modulate Nrf1 and/or Nrf2 activity.

The impaired activation of Nrf2 and the resulting elevated ROS levels were found to be essential for the HCV life cycle. Increased ROS levels further result in the induction of the autophagic pathway, which plays an essential role in the MVB-dependent release of HCV particles. Nevertheless, the oxidative stress further contributes to the HCV-associated pathogenesis, as elevated ROS levels induce DNA damage, which in turn leads to genetic mutations of the host genome. In addition, impaired Nrf2-functionality has been reported to interfere with liver regeneration based on a ROS-mediated inhibition of the insulin/insulin growth factor 1 (IGF1) signalling cascade [45,55]. Another factor involved in redox homeostasis is the transcription factor Nrf1. However, besides its role in redox homeostasis, Nrf1 is implicated in maintaining lipid and cholesterol homeostasis and modulates inflammatory processes. In this study, we were curious if the inhibition of Nrf1-dependent transcriptional activation, caused by HCV, could affect lipid metabolism and thereby impact viral replication, as the HCV life cycle is tightly connected to lipid metabolism. One important factor involved in the removal of intracellular cholesterol is the liver X receptor alpha (LXR-α). Notably, Nrf1 interferes with LXR activities to support cholesterol homeostasis [29]. In the context of HCV, the LXR agonists GW3965 and T0901317 or the natural LXR ligand 24(S),25-epoxycholesterol have been reported to inhibit HCV infection [56]. In accordance with these reports, we could observe that in HCV-positive cells, the activation of the LXRα promoter is impaired. It can be assumed that the defect in the Nrf1-LXRα-axis in HCV-positive cells, along with other factors, further results in elevated intracellular cholesterol levels. Yet, this phenotype can be partially rescued by co-expression of the 85 kDa Nrf1 isoform, corroborating the essential role of Nrf1 in maintaining lipid homeostasis. Vice versa, inhibition of Nrf1 activity by co-expression of the 25 kDa form of Nrf1 in HCV-negative cells leads to elevated intracellular cholesterol levels. Interestingly, a recent study described the interplay of Nrf1 and Nrf2 in the modulation against excess liver cholesterol [52].

Cholesterol-rich membrane domains play a central role in the release and infectivity of progeny virus, virus entry, and replication, i.e., affecting the ‘membranous web’ formation [57,58,59]. In healthy adipocytes, lipid droplets serve as the main storage organelle for free cholesterol and its esterified derivatives [60,61]. In the case of HCV infection, intracellular accumulation of lipids in the form of lipid droplets is a prerequisite for viral replication. LDs play an important role as the site of viral morphogenesis and as a central part of the ‘membranous web’ [50]. Furthermore, in HCV-positive cells, activation of LXRα is impaired, preventing the induction of an effective cholesterol removal programme. This leads to elevated intracellular cholesterol levels, which favour HCV replication and can contribute to HCV-associated pathogenesis as steatosis [62]. Apart from these direct effects of Nrf1 on lipid metabolism and thereby on the HCV life cycle, there might be a variety of indirect factors involved. Several kinome-based studies performed by our group yielded a fundamental insight into the host–virus interplay [63,64,65,66]. Therefore, we decided to utilise this tool in the context of HCV-Nrf1 interplay. We aimed to identify a general trend in the deregulation of kinases that could potentially serve as drug targets against viral infection. We were able to pinpoint a set of kinases involved in inflammatory processes and, most importantly, cholesterol metabolism. The most prominent example is PTK6, as it is the kinase that appears in both HCV-positive cells and HCV-negative cells expressing the 25 kDa Nrf1 isoform. PTK6 is described to modulate autophagy pathways; thus, it potentially plays a role in HCV pathogenesis, as HCV is strongly dependent on autophagy [67]. Moreover, the analysis revealed that overexpression of the Nrf1-85kDa fragment triggers an activation of AMP-activated kinase (AMPK). It has recently been described that NS5A has the capacity to inactivate AMPK. This leads to a diminished nuclear import of the RNA-binding protein HuR through the inhibition of AMPK-mediated phosphorylation and acetylation of importin-α1. Cytoplasmic HuR is crucial for HCV replication. It is involved in the assembly of the replication complex on \ viral-3′UTR, and the loss of cytoplasmic HuR hampers viral replication. The activation of AMPK decreases cytoplasmic HuR and thereby reduces HCV replication [68]. Moreover, the strong impact of trans-dominant-negative Nrf1 (25 kDa form) in GND cells on the inhibitor of the nuclear factor kappa B kinase subunit epsilon (IKBKE) reflects a potent role in the regulation of inflammatory processes and underlines the potential relevance for HCV-associated pathogenesis. This indicates that, in addition to the direct impact of Nrf1 on the intracellular lipid content, a variety of further factors are relevant for both HCV replication and HCV-associated pathogenesis. The kinome analysis and kinases that we were able to identify could serve as an anchoring point for more extended and detailed studies in the future, identifying therapeutic targets in real-life clinical settings.

Taken together, these data describe crosstalk between HCV and Nrf1. In light of Nrf1’s prominent role in controlling intracellular ROS levels, expressing cytoprotective genes, and modulating lipid metabolism and inflammatory processes, as well as the impact of deregulated ROS levels and lipids on the HCV life cycle and HCV-associated pathogenesis, our data might contribute to our understanding of HCV-associated pathogenesis and the identification of therapeutic intervention targets.

There is a correlation between the HCV-dependent deregulation of Nrf1-driven transcriptional activation and the HCV life cycle. One of the underlying mechanisms is the modulation of the Nrf1-sMaf interaction, which affects, inter alia, the expression of ARE-regulated genes. Moreover, disrupted Nrf1 function impairs lipid removal programmes, leads to elevated intracellular cholesterol content, and influences the number and volume of lipid droplets, thus favouring the HCV life cycle. Apart from the impact on the HCV life cycle, the inflammatory processes are deregulated once again, reflecting the relevance of Nrf1 for HCV-associated pathogenesis. It is tempting to speculate if reconstitution of Nrf1 activity could represent a new therapeutic target by initiating a cholesterol removal programme affecting HCV replication and, in consequence, HCV-associated pathogenesis. Future studies will further characterise the role of various Nrf1 isoforms in HCV-associated infection. Particular emphasis will be put on lipid metabolism, as it is a central factor in viral morphogenesis, and the full-length Nrf1, as it encompasses the cholesterol-binding domain.

## Figures and Tables

**Figure 1 viruses-17-01052-f001:**
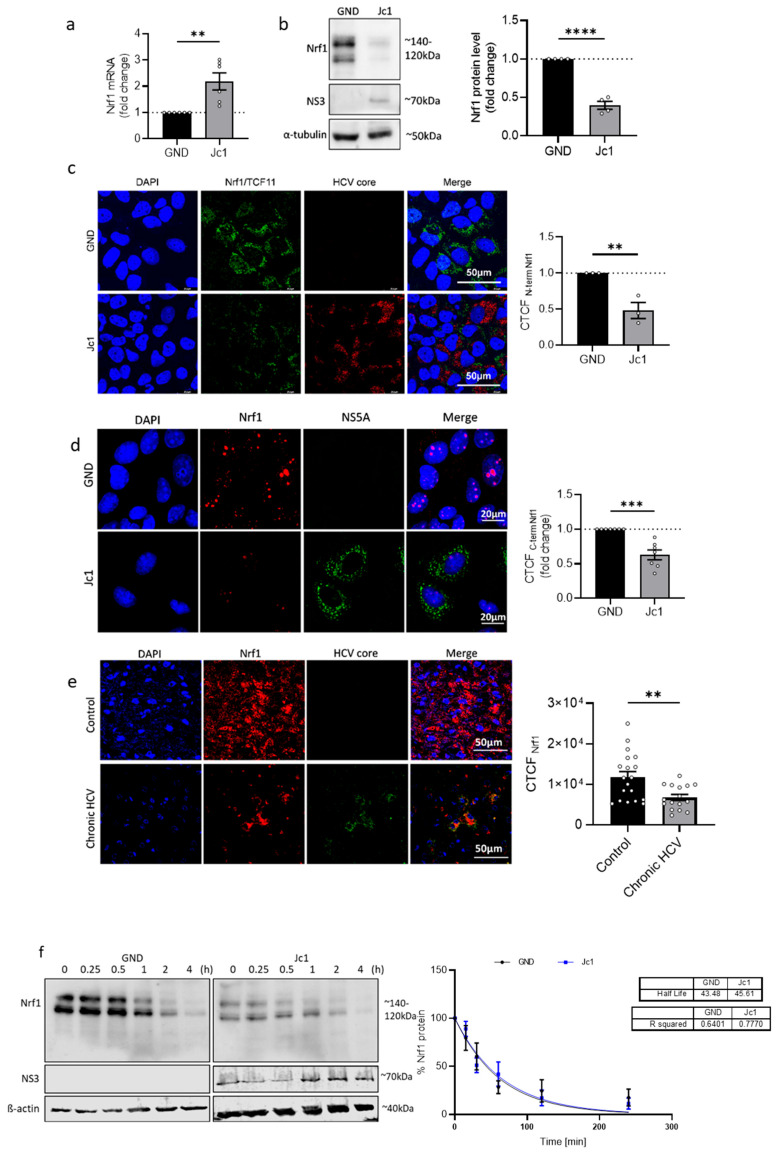
Decreased Nrf1 amount in HCV-replicating cells. (**a**) qPCR analysis to monitor the level of Nrf1-specific transcripts in stably HCV-replicating (Jc1) Huh7.5 cells and the corresponding control cells (GND) 72 h post seeding. Relative values of the Nrf1 mRNA levels referred to the GND cells (set to 1); *n* = 6 biological replicates. (**b**) Representative Western blot and the respective densitometric analyses of cellular lysates derived from stably HCV-replicating (Jc1) Huh7.5 cells and the GND control cells. Nrf1 was detected using an Nrf1-specific antibody binding to the N-terminal part of the protein. In addition, NS3 was detected to confirm HCV replication. Detection of alpha-tubulin served as loading control. Relative values referred to the control cells (GND) (set to 1); *n* = 4 biological replicates. (**c**) CLSM analysis of stably HCV-replicating (Jc1) Huh7.5 cells and the corresponding control (GND) cells. For the detection of Nrf1, an antibody binding to the N-terminal part of the protein was used (green). HCV-positive cells were detected using an NS3-specific antibody (red). Nuclei were stained using DAPI (blue). Scale bar: 50 µM. Quantification of the relative CTCF (corrected total cell fluorescence) of Nrf1. Relative values referred to control cells (GND) (set to 1). For each condition, a minimum of 10 cells was analysed. (**d**) CLSM analysis of stably HCV-replicating (Jc1) Huh7.5 cells and the corresponding control (GND) cells. The cells were fixed using a 1:1 ethanol–acetone mixture for 10 min. For the detection of Nrf1, an antibody binding to the C-terminal part was used (red), and HCV-positive cells were visualised using an NS5A-specific antibody (green). Nuclei were stained using DAPI (blue). Quantification of Nrf1-specific signal intensity in the nucleus is expressed as relative CTCF. Relative values referred to the control cells (GND) (set to 1). (**e**) CLSM analysis of liver tissue derived from a patient suffering from chronic HCV or a non-infected patient. For the detection of Nrf1, an Nrf1-specific antibody binding to the N-terminal part was used (red). HCV-positive cells were visualised using an HCV core-specific antibody (green). Nuclei were stained using DAPI (blue). Scale bar: 50 µm. Quantification of the Nrf1-specific signal intensity is expressed as CTCF. For each condition, a minimum of 10 cells was analysed. (**f**) Representative Western blot and the respective densitometric analysis of cellular lysates derived from HCV-positive (Jc1) and corresponding negative (GND) cells. To determine the half-life of Nrf1, the cells were treated with 142 μM cycloheximide (CHX) for the indicated time points (0 to 240 min). Nrf1-specific fragments were detected using an Nrf1-specific antibody binding to the N-terminal part of the protein. In addition, NS3 was detected using an NS3-specific antibody to confirm HCV replication. Detection of beta-actin served as loading control. Relative change in Nrf1 signal intensity expressed as % of signal intensity at 0 min CHX-treatment. Curve fitting was applied as a one-phase decay model with an intercept set to 100 and decay set to reach 0%. *n* = 3 biological replicates. (**a**–**f**) For all experiments, statistics were performed as mean ± SEM, and unpaired *t*-test referred to control; ns—not significant, ** *p* < 0.01, *** *p* < 0.001, **** *p* < 0.0001.

**Figure 2 viruses-17-01052-f002:**
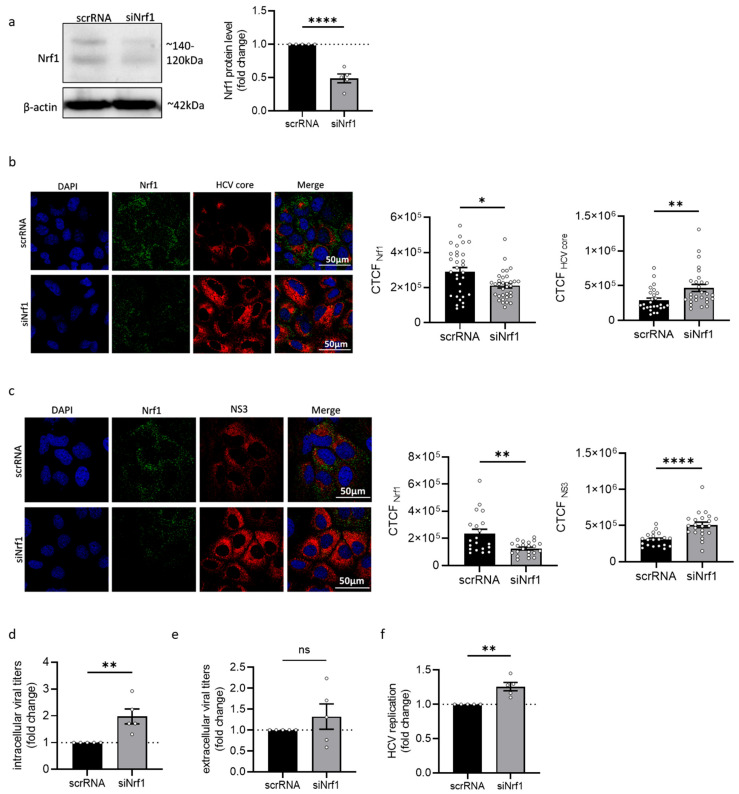
Silencing of Nrf1 leads to increased HCV replication. (**a**) Representative Western blot and respective densitometric analysis of cellular lysates derived from stably HCV-replicating (Jc1) Huh7.5 cells 96 h post-transfection. To silence Nrf1 expression, cells were transfected with Nrf1-specific siRNA or scrRNA as a control. Nrf1 was detected using an Nrf1-specific antibody binding to the N-terminal part of the protein. Detection of beta-actin served as loading control. Relative values referred to the control cells transfected with scrRNA (set to 1). *n* = 4. (**b**,**c**) CLSM analysis of stably HCV-replicating (Jc1) Huh7.5 cells with silenced Nrf1 expression (siNrf1) and the corresponding control cells (scrRNA). The cells were fixed with 4% PFA 96 h post-transfection. For the detection of Nrf1, an antibody binding to the N-terminal part was used (green). HCV core or NS3 were detected using specific antibodies (red). Nuclei were stained with DAPI (blue). Scale bar: 50 µm. Quantification of Nrf1-specific, HCV core-specific or NS3-specific signal intensity is expressed as CTCF. For each condition, a minimum of 10 cells was analysed. (**d**,**e**) Detection of intra- and extracellular viral particles by TCID50 96 h after silencing Nrf1. Relative values referred to the control cells transfected with scrRNA (set to 1). *n* = 5. (**f**) Luciferase reporter gene assay of Huh7.5 cells replicating a HCV-luc reporter construct with silenced Nrf1 expression (siNrf1) and the corresponding control cells (scrRNA). Relative values referred to the control cells transfected with scrRNA (set to 1). *n* = 5. For all experiments, statistics were performed as mean ± SEM, and unpaired *t*-test or Mann–Whitney test referred to control; ns—not significant, * *p* < 0.05, ** *p* < 0.01, **** *p* < 0.0001.

**Figure 3 viruses-17-01052-f003:**
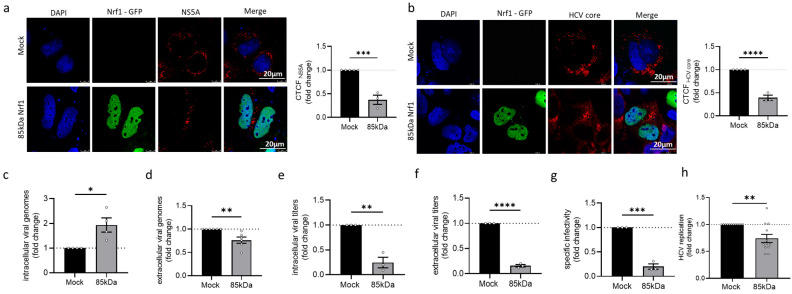
Overexpression of Nrf1 restricts HCV. Stably HCV-replicating Huh7.5 cells (Jc1) were transfected with the Nrf1-85kDa-GFP construct or the empty vector (pJo23) (Mock) for 48 h. (**a**,**b**) CLSM analysis. The cells were fixed with 4% PFA. For visualisation of 85 kDa Nrf1, the GFP-specific fluorescence was detected (green). (**a**) NS5A and (**c**) HCV core were visualised using specific antibodies (red). Nuclei were stained with DAPI (blue). Scale bar: 20 µm. Quantification of NS5A-specific and core-specific signal intensity is expressed as CTCF. Relative values referred to the control cells (GND) (set to 1). For each condition, a minimum of 10 cells was analysed. (**c**,**d**) qPCR analyses to monitor (**c**) intracellular and (**d**) extracellular viral genomes. Values referred to the Mock-transfected cells (set to 1). *n* = 4 (intracellular viral RNA) and *n* = 5 (extracellular viral RNA). (**e**,**f**) Detection of intra- and extracellular viral particles by TCID_50._ Relative values referred to the Mock-transfected cells (set to 1). *n* = 3. (**g**) The specific infectivity after Nrf1-85kDa-GFP overexpression was assessed by calculating the ratio of extracellular infectious viral particles (TCID_50_/_mL_), determined by TCID_50_, the total amount of extracellular viral genomes (genomes/mL), and qPCR. Relative values referred to the Mock-transfected cells (set to 1). *n* = 3. (**h**) Luciferase reporter gene assay of Huh7.5 cells replicating a HCV-luc reporter construct and the corresponding control cells. Relative values referred to the Mock-transfected cells (set to 1). *n* = 5. For all experiments, statistics were performed as mean ± SEM, and unpaired *t*-test referred to control; ns—not significant, * *p* < 0.05, ** *p* < 0.01, *** *p* < 0.001, **** *p* < 0.0001.

**Figure 4 viruses-17-01052-f004:**
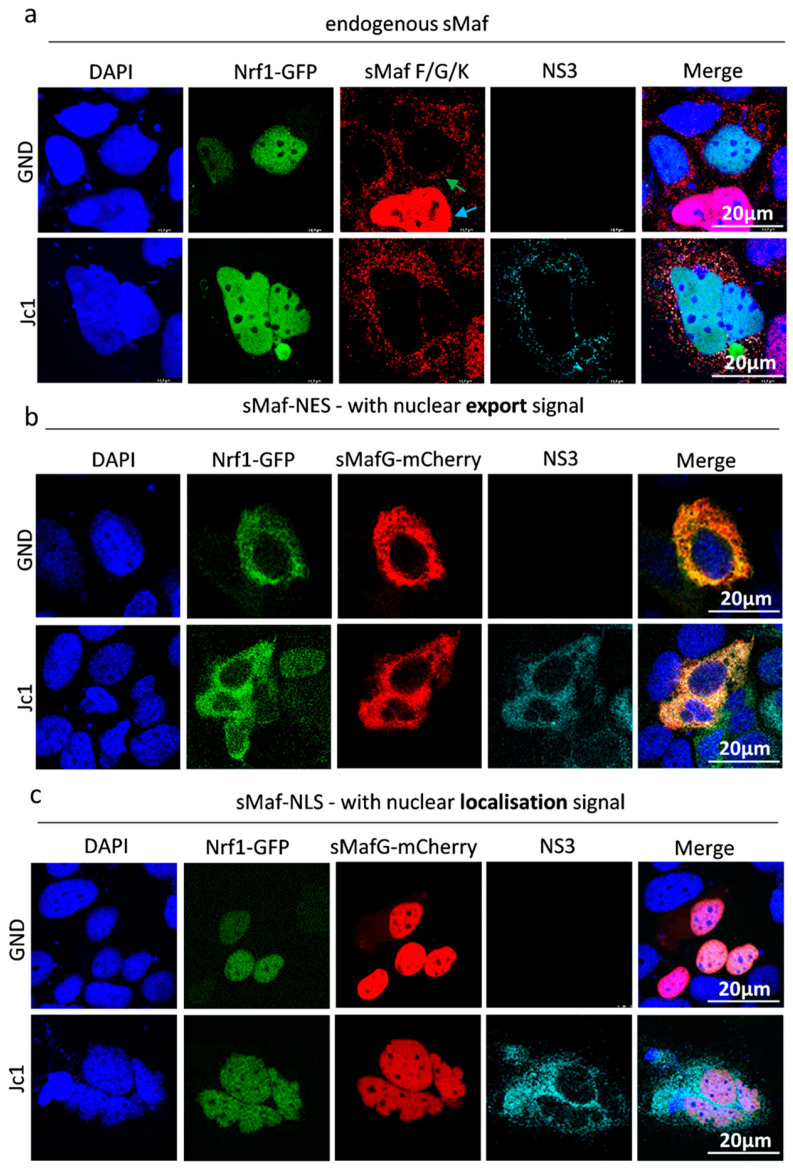
Delocalisation of Nrf1 by sMaf-NES/NLS fusion proteins. CLSM analysis. Stably HCV-replicating Huh7.5 cells (Jc1) or corresponding control (GND) cells were transfected with the Nrf1-85kDa-GFP construct for 48 h. (**a**) The cells were co-transfected with an empty vector (pJo23), (**b**) a vector encoding sMaf-NES-mCherry fusion protein, or (**c**) a vector encoding sMaf-NLS-mCherry fusion protein. The cells were fixed with 4% PFA. For the visualisation of 85 kDa Nrf1, the GFP-specific fluorescence was detected (green). For the detection of endogenous sMaf proteins, a specific antibody was used (red). For the visualisation of sMaf with a nuclear localisation/export signal, the mCherry-specific fluorescence was detected. HCV-positive cells were visualised using an NS3-specific antibody (cyan). Nuclei were stained using DAPI (blue). Scale bar: 20 µm. Images are representative of 3 biological replicates.

**Figure 5 viruses-17-01052-f005:**
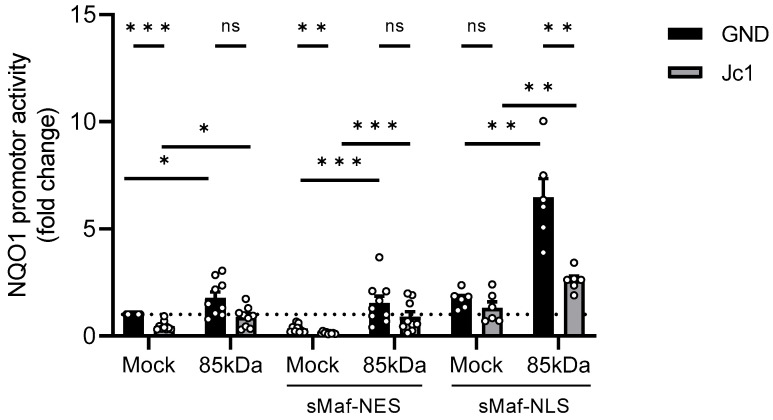
Impaired activation of Nrf1/ARE-dependent gene expression after overexpression of sMaf-NES/NLS. Stably HCV-replicating Huh7.5 cells (Jc1) or corresponding control (GND) cells were transfected with the Nrf1-85kDa-GFP construct or an empty vector (Mock) (pJo23) for 48 h. The cells were co-transfected with a reporter construct expressing the luciferase gene under control of the NQO1 promoter and an empty vector (pUc18), with a vector encoding sMaf-NES-mCherry fusion protein, or with a vector encoding sMaf-NLS-mCherry fusion protein. Relative luciferase activity for Mock-transfected cells was arbitrarily set to 1, as visualised by the dotted line. *n* = 9 (pUc18 and sMaf-NES) and *n* = 6 (sMaf-NLS). For all experiments, statistics were performed as mean ± SEM, and unpaired *t*-test or Mann–Whitney test referred to control; ns—not significant, * *p* < 0.05, ** *p* < 0.01, *** *p* < 0.001.

**Figure 6 viruses-17-01052-f006:**
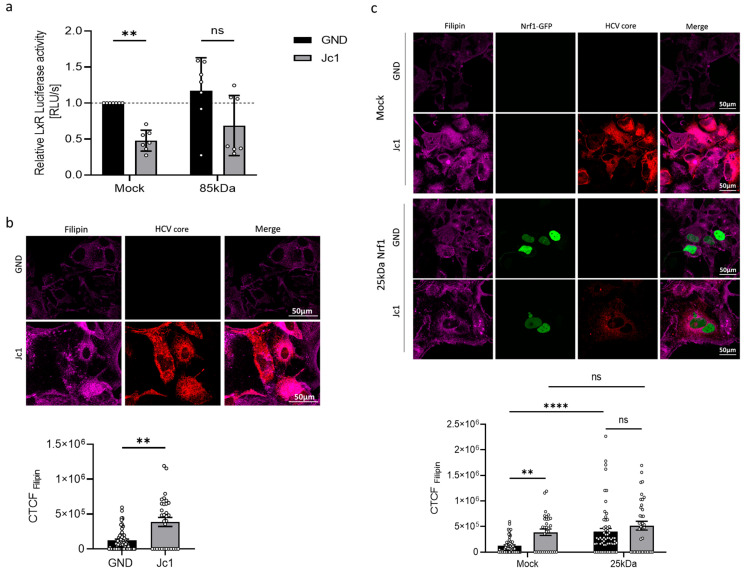
HCV impairs activation of LXR promoter. (**a**) Stably HCV-replicating Huh7.5 cells (Jc1) or corresponding control (GND) cells were transfected with the Nrf1-85kDa-GFP construct or an empty vector (Mock) (pJo23) for 48 h. The cells were co-transfected with a reporter construct expressing the luciferase gene under control of the LxR-α promoter. Relative luciferase activity for Mock-transfected cells was arbitrarily set to 1 as visualised by the dotted line. *n* = 7. (**b**) CLSM analysis. The cells were fixed with 4% PFA. HCV-positive cells were visualised using anti-HCV core-specific antibody (red). Intracellular cholesterol level was visualised with filipin III (magenta). Quantification of the filipin-specific signal intensity is expressed as CTCF. Scale bar: 50 µm. Relative values referred to Mock-transfected cells (set to 1). For each condition, a minimum of 10 cells was analysed. (**c**) CLSM analysis. Stably HCV-replicating Huh7.5 cells (Jc1) or corresponding control (GND) cells were transfected with the Nrf1-25kDa-GFP construct or an empty vector (Mock) (pJo23) for 48 h. The cells were fixed with 4% PFA. For visualisation of 25 kDa Nrf1, the GFP-specific fluorescence was detected (green). HCV-positive cells were visualised using anti-HCV core-specific antibody (red). Intracellular cholesterol level was visualised with filipin III (magenta). Quantification of the filipin-specific signal intensity is expressed as CTCF. Scale bar: 50 µm. For each condition, a minimum of 10 cells was analysed. For all experiments, statistics were performed as mean ± SEM, and Mann–Whitney test referred to control; ns—not significant, ** *p* < 0.01, **** *p* < 0.0001.

**Figure 7 viruses-17-01052-f007:**
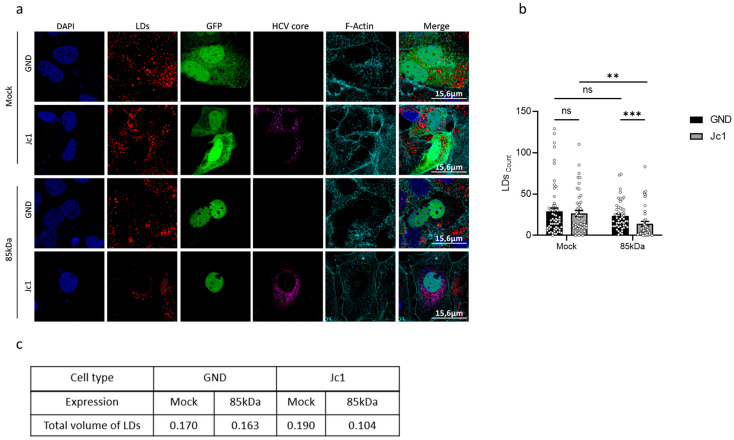
The number of liquid droplets and the total volume are affected by Nrf1 activity. (**a**) CLSM analysis. Stably HCV-replicating Huh7.5 cells (Jc1) or corresponding control (GND) cells were transfected with the Nrf1-85kDa-GFP construct or with a vector expressing EGFP (Mock) for 48 h. For 85 kDa Nrf1 visualisation, GFP-specific signal was detected (green). HCV core was detected using a specific antibody (magenta). Lipid droplets were stained with Nile Red (red). F-actin was stained with phalloidin-Atto 633 (cyan). Scale bar: 15.6 µm. Images are representative of 3 biological replicates. The number (**b**) was determined using Fiji (Image J) software. A minimum of 10 cells was analysed. For all experiments, statistics were performed as mean ± SEM, and Mann–Whitney test referred to control; ns—not significant, ** *p* < 0.01, *** *p* < 0.001. The total volume of lipid droplets (**c**) was calculated based on the average number of LDs multiplied by the average volume of LDs. For the calculation, the number and perimeter were determined using Fiji (Image J) software. For each condition, a minimum of 10 cells was analysed.

**Figure 8 viruses-17-01052-f008:**
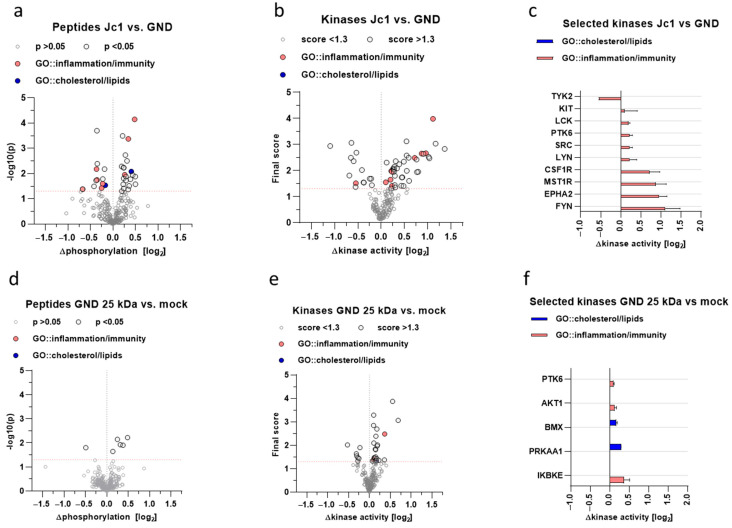
Inhibitory fragments of Nrf1 modulate the host kinome related to inflammation, innate immunity, and lipid metabolism. Kinome profiling of stably HCV-replicating Huh7.5 cells (Jc1) or corresponding control (GND) cells transfected with a vector overexpressing the 25 kDa fragment of Nrf1 or an empty vector (Mock) (pJo23). (**a**,**b**) Volcano plots of differential peptide phosphorylation in cell lysates of Mock-transfected Jc1 versus GND cells or GND cells overexpressing 25 kDa Nrf1 versus Mock-transfected GND cells; each dot represents a distinct 13-mer peptide derived from host proteins; values on x-axis displayed in log2-space; values on y-axis reflect significance; significance cut-off set to *p* < 0.05 as indicated by dashed, red line. (**c**,**d**) Volcano plots of predicted differential kinase activity based on the phosphorylation pattern in A-B; values on x-axis displayed in log2-space; each dot represents a distinct kinase; values on y-axis reflect the final score of predicted kinases; significance cut-off set to score < 1.3 as indicated by dashed, red line. (**e**,**f**) Detailed depiction of kinases and their activity marked in C-D; values depicted as mean −/+SD. Grey and black colouring represent peptides or kinases below or above threshold, respectively; red or blue colouring represent peptides or kinases being part of the following gene ontology terms: inflammatory response (GO:0006954) and innate immune response (GO:0045087), or cholesterol biosynthetic process (GO:0006695) and lipid biosynthetic process (GO:0008610), respectively.

## Data Availability

The datasets generated during and/or analysed during the current study are available from the corresponding author on reasonable request.

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
