# Peer review of "The HCV-Dependent Inhibition of Nrf1/ARE-Mediated Gene Expression Favours Viral Morphogenesis"

_viruses, 2025, doi:10.3390/v17081052_

Round 1

Reviewer 1 Report

Comments and Suggestions for Authors

In this manuscript, Szostek and colleagues investigated the relationship between HCV infection and nuclear factor erythroid 2 related factor 1 (Nrf1). Nrf1 has numerous cellular roles including responding to oxidative stress and excess cholesterol. The authors observe that HCV infection increases Nrf1 mRNA levels but decreases its protein expression, although the half-life of Nrf1 is consistent in mock or HCV infection. Transient knockdown of Nrf1 expression led to slightly increased HCV infection, marked by an increase in core/NS3 expression and HCV replication. Conversely, overexpression of the active (85 kDa) form of Nrf1 resulted in decreased viral protein expression, titer and specific infectivity. Since Nrf1/2 have been demonstrated in literature to interact with sMaf transcription factors, the authors investigated the localization of sMafs following overexpression of 85 kDa Nrf1. Nrf1 overexpression resulted in decreased nuclear localization of sMafs, likely resulting in impaired activation of Nrf1/ARE-dependent gene expression. The authors next tested the effect of HCV infection and Nrf1 on LXR-a, a factor involved in regulating cholesterol levels. HCV infection was found to decrease LXR promoter activity and increase intracellular cholesterol levels. Similarly, overexpression of the dominant negative (25 kDa) Nrf1 isoform also increased intracellular cholesterol levels. Thus, downregulation of Nrf1 by HCV may contribute to the observed changes to cholesterol levels. To further investigate the impact on cholesterol and lipid metabolism, Szostek et al assessed the effect of altered Nrf1 activity on lipid droplets. Overexpression of active (85 kDa) Nrf1 reduced the total number and volume of lipid droplets in HCV-infected cells. Lastly, using kinome analysis the authors show that inhibition of Nrf1 alters the phosphorylation of innate immune and lipid metabolism related proteins.

Overall, the authors have provided new insight into the regulation of lipid metabolism in HCV-infected cells. The study is nicely done, with data presented that are clear and appropriately interpreted, and for the most part the manuscript is well-written. The manuscript would be further strengthened by some minor clarifications/considerations.

Minor comments:

  1. More context on the chronic HCV infected liver samples should be provided in the text (e.g. HCV genotype). Presumably there was ethics approval for the collection and use of these human samples which should be stated in the text.
  2. The authors frequently use the term “replicon” to refer to the HCV replicase complex. For clarity, it would be helpful to refer to this as the replicase complex rather than replicon (to avoid confusion, since the authors are using HCV infection models rather than replicon models in the manuscript).
  3. On Line 445, the authors conclude that “these data indicate that the 85 kDa Nrf1 has the capacity to bind to sMaf proteins”. While it is clear that Nrf1 induces relocalization of sMaf, the data do not show evidence of direct binding. As such, this reviewer suggests either evaluating binding of Nrf1 to sMaf proteins to support this conclusion, or toning down the statement.
  4. Section 3.6 would be strengthened by explaining the relationship between LXR-a and Nrf1 and the rationale for specifically focusing on LXR-a.
  5. For Figure 7C, it would be helpful to indicate in the chart itself (not just the caption) that the numbers refer to LD volume.
  6. Inclusion of a graphical abstract is appreciated. However, some of the text in the graphic is quite small and difficult to read, and organelles such as the nucleus and ER should be labelled accordingly for clarity.

Author Response

Reviewer #1: 
1. More context on the chronic HCV infected liver samples should be provided in the text (e.g. HCV genotype). Presumably there was ethics approval for the collection and use of these human samples which should be stated in the text.

As requested by the reviewer we  we provided more information about the patients and the HCV genotype they were infected with. This information is provided on lines 218-222 of the revised manuscript: “Immunostaining procedure was performed on paraffin-embedded human liver sections, obtained from patients chronically infected with HCV genotype 1a and healthy individuals as controls. Both infected and healthy individuals were male, aged between 45 and 58. The specimens were kindly provided by K. Klingel, Institute of Molecular Pathology, Universitätsklinikum Tübingen, Germany. Samples collection was supervised by the Ethics Comitee at the Medical Faculty of the Eberhard Karls University and at the University Hospital of Tübingen

2. The authors frequently use the term “replicon” to refer to the HCV replicase complex. For clarity, it would be helpful to refer to this as the replicase complex rather than replicon (to avoid confusion, since the authors are using HCV infection models rather than replicon models in the manuscript).

As suggested by the reviewer, for better clarity the term “replicon complex” was rephrased and changed to  “replicase complex”. 

3. On Line 445, the authors conclude that “these data indicate that the 85 kDa Nrf1 has the capacity to bind to sMaf proteins”. While it is clear that Nrf1 induces relocalization of sMaf, the data do not show evidence of direct binding. As such, this reviewer suggests either evaluating binding of Nrf1 to sMaf proteins to support this conclusion, or toning down the statement.

We agree with the reviewer that our data do not fully corroborate  the conclusion of a direct binding of sMaf to Nrf1. However, the 10 day revision time is not sufficient for conducting experiments evaluating the direct binding of Nrf1 to sMaf in detail. Yet according to the reviewer’s suggestion we modified the aforementioned conclusion accordingly.
Lines 454-455  of the revised manuscript:“These data indicate that the 85 kDa Nrf1 has the capacity to interact with sMaf proteins and thus can be withdrawn from the nucleus by delocalized sMaf proteins in HCV-positive and HCV-negative cells”.

4. Section 3.6 would be strengthened by explaining the relationship between LXR-a and Nrf1 and the rationale for specifically focusing on LXR-a.

We see the point raised by the reviewer. To make the rationale of this experiment clearer, we have prefaced this section with an additional explanation in the revised version of the manuscript. 
Lines 497-502 of the revised manuscript: “According to a model formulated by Widenmaier et al. [28] Nrf1 is likely to be involved in regulation of the LXR activity. Therefore, we decided to investigate whether overexpression of the transciptionally active Nrf1 form, the 85 kDa fragment, would rescue LXR promoter activity in HCV-positive cells. Surprisingly, overexpression of 85 kDa Nrf1 only slightly increased the activation of the LXR-α promoter in HCV-positive as well as HCV-negative cells. Therefore, the direct relevance of Nrf1 for the HCV-dependent inhibition of LxR promoter remains open (Figure 6a)”.

5. For Figure 7C, it would be helpful to indicate in the chart itself (not just the caption) that the numbers refer to LD volume.

Following the reviewer’s suggestion, the content of chart in figure 7c has been described in more detail. (Lines 552-554 of the revised manuscript).

6. Inclusion of a graphical abstract is appreciated. However, some of the text in the graphic is quite small and difficult to read, and organelles such as the nucleus and ER should be labelled accordingly for clarity.

We agree that some parts of the graphical abstract might be difficult to read, therefore we performed the requested necessary corrections. The important organelles – nucleus and ER – are also labeled on the scheme (top of page 2 of the revised manuscript). 

Reviewer 2 Report

Comments and Suggestions for Authors

I have carefully analyzed the manuscript as a whole and also directed to the specific sections from the article: „The HCV-dependent inhibition of Nrf1/ARE-mediated gene expression favours viral morphogenesis” by Olga Szostek , Patrycfja Schorsch , Daniela Bender , Mirco Glitscher , Eberhard Hildt.

I observed that the results of the paper are extremely interesting, the novelty of the results is very good, the article is detailed and accurate, and the paper is suitable for the journal’s aims.

Also, the subject is highly interesting: Nrf1 is a key transcription factor that plays a role in regulating genes involved in antioxidant activity, detoxification, and proteasome function.

HCV infection can decrease the function and synthesis of Nrf1-regulated genes, increasing the oxidative stress and contributing to liver damage and HCV induced liver disease progression. 

My suggestions are as follows:

In the Introduction part, a small paragraph related to the implication of chronic hepatitis C infection as an important public health problem worldwide is needed in my opinion.

The Methods part is detailed and accurate, all the methods are well explained.

In the Results part:

At line 410 - If the overexpression of the 85 kDa Nrf1-specific fragments impairs the formation of intact viral particles and their release – is it possible to use this information to target the replication in patients with acute HCV infection in order to decrease the high rate of chronic HCV infections (75-80%)?

In the last part of the article the significant outcome of the study needs to be more specifically underlined.  Can some of these results be used as a marker of evolution of the chronic HCV liver induced disease progression in the real-life clinical settings?

The English language is very good and it does not need to be improved.

Still, in my opinion the results seem to be very similar with the results and methods used in the PhD thesis of Patrycfja Schorsch, conducted by Prof. Eberhard Hildt at Paul Ehrlich Institute in 2024.

Author Response

Reviewer #2: 
1. In the Introduction part, a small paragraph related to the implication of chronic hepatitis C infection as an important public health problem worldwide is needed in my opinion.
As requested by the reviewer we included this  important information in the revised manuscript. Therefore a paragraph describing HCV infection implications is included in the introduction. (Lines 35-43 of the revised manuscript
 “It is estimated, that more than 50 million people globally are suffering from hepatitis C infection. On average, 30% of infected individuals experience the acute form of viral hepatitis, which in most cases can be cleared spontaneously. In the remaining 70% of the cases HCV infection leads to a chronic viral hepatitis C (CHC). CHC is a significant health burden worldwide, with serious implications such as liver cirrhosis, hepatocellular carcinoma, increased risk of type 2 diabetes or hypercholesterolemia [3–5]. Current therapeutic approaches against HCV infection involve the use of third-generation direct-acting antivirals (DAAs), which have replaced the earlier interferon-alpha (IFN-α) and Ribavirin-based treatments [6]. Nevertheless, due to economical inequity in low-income countries often the accessibility to DAAs remains poor alongside with social awareness [7]. Therefore, the development of an effective HCV vaccine remains an ongoing challenge [8]”.
2. The Methods part is detailed and accurate, all the methods are well explained.
Thank You!
In the Results part:
3. At line 410 - If the overexpression of the 85 kDa Nrf1-specific fragments impairs the formation of intact viral particles and their release – is it possible to use this information to target the replication in patients with acute HCV infection in order to decrease the high rate of chronic HCV infections (75-80%)?
We agree with the reviewer that this could be an interesting approach to control HCV replication. However, at our current stage of knowledge there a variety of further studies might be required to deepen our knowledge about the HCV-Nrf1-crosstalk and its potential relevance for antiviral strategies. Yet it is tempting to speculate and encouraging that our studies can serve as the base for further clinical studies.   
4. In the last part of the article the significant outcome of the study needs to be more specifically underlined.  Can some of these results be used as a marker of evolution of the chronic HCV liver induced disease progression in the real-life clinical settings?
The significance of the study was highlighted and emphasised in the discussion part of the revised version of the manuscript in lines 702-703 “We aimed to identify a general trend in the deregulation of kinases which could potentially serve as drug targets against viral infection”; Moreover, we added a statement summarizing the relevance of our observations for a deeper understanding of HCV-associated pathogenesis and HCV life cycle as a base for novel intervention strategies. 
Lines 717-723 of the revised manuscript: “The kinome analysis and kinases that we were able to identify could serve as an anchoring point for future more extedned and detailed studies for identifying therapeutic targets in real-life clinical settings.
Taken together, these data describe crosstalk between HCV and Nrf1. In light of Nrf1's prominent role in controlling intracellular ROS levels, expressing cytoprotective genes, and modulating lipid metabolism and inflammatory processes, as well as the impact of deregulated ROS levels and lipids on the HCV life cycle and HCV-associated pathogenesis, our data might contribute to our understanding of HCV-associated pathogenesis and the identification of therapeutic intervention targets.”

5. The English language is very good and it does not need to be improved.
Thank You!
6. Still, in my opinion the results seem to be very similar with the results and methods used in the PhD thesis of Patrycfja Schorsch, conducted by Prof. Eberhard Hildt at Paul Ehrlich Institute in 2024.
That is correct. The manuscript contains data gathered by Patrycja Schorsch during her doctoral studies at Paul-Ehrlich-Institute. Yet, the results presented in the manuscript as well as in the thesis have not been published so far. 

Round 2

Reviewer 2 Report

Comments and Suggestions for Authors

The authors have properly addressed all the suggestions and comments, indicating a positive response to feedback, I recommend the acceptance.